# Identification of the *Telomere elongation* Mutation in *Drosophila*

**DOI:** 10.3390/cells11213484

**Published:** 2022-11-03

**Authors:** Hemakumar M. Reddy, Thomas A. Randall, Francesca Cipressa, Antonella Porrazzo, Giovanni Cenci, Radmila Capkova Frydrychova, James M. Mason

**Affiliations:** 1Laboratory of Molecular Genetics, National Institute of Environmental Health Sciences, Research Triangle Park, NC 27713, USA; 2Nabsys 2.0, LLC, 60 Clifford Street, Providence, RI 02903, USA; 3Integrative Bioinformatics, National Institute of Environmental Health Sciences, Research Triangle Park, NC 27713, USA; 4Department of Ecological and Biological Sciences, Università degli Studi della Tuscia, 01100 Viterbo, Italy; 5Dipartimento di Biologia e Biotecnologie “C. Darwin”, Sapienza Università di Roma, 00185 Rome, Italy; 6Unit of Molecular Genetics of Complex Phenotypes, Bambino Gesù Children’s Hospital, IRCSS, 00146 Rome, Italy; 7Fondazione Cenci Bolognetti/Istituto Pasteur Italia, 00185 Rome, Italy; 8Institute of Entomology, Biology Centre AS CR, v.v.i., 370 05 Ceske Budejovice, Czech Republic; 9Faculty of Science, University of South Bohemia, 370 05 Ceske Budejovice, Czech Republic

**Keywords:** *Drosophila melanogaster*, telomere, next-generation sequencing, transposon-induced recombination

## Abstract

Telomeres in *Drosophila melanogaster*, which have inspired a large part of Sergio Pimpinelli work, are similar to those of other eukaryotes in terms of their function. Yet, their length maintenance relies on the transposition of the specialized retrotransposons *Het-A*, *TART*, and *TAHRE*, rather than on the activity of the enzyme telomerase as it occurs in most other eukaryotic organisms. The length of the telomeres in *Drosophila* thus depends on the number of copies of these transposable elements. Our previous work has led to the isolation of a dominant mutation, *Tel^1^*, that caused a several-fold elongation of telomeres. In this study, we molecularly identified the *Tel^1^* mutation by a combination of transposon-induced, site-specific recombination and next-generation sequencing. Recombination located *Tel^1^* to a 15 kb region in 92A. Comparison of the DNA sequence in this region with the Drosophila Genetic Reference Panel of wild-type genomic sequences delimited *Tel^1^* to a 3 bp deletion inside intron 8 of *Ino80.* Furthermore, CRISPR/Cas9-induced deletions surrounding the same region exhibited the *Tel^1^* telomere phenotype, confirming a strict requirement of this intron 8 gene sequence for a proper regulation of *Drosophila* telomere length.

## 1. Introduction

Telomeres in all eukaryotes are functionally similar, although structural differences between some species exist [1]. Linear chromosome ends are not replicated completely, and telomeres must compensate this end replication problem by adding new sequences at the chromosome end. The majority of eukaryotes use a specialized reverse transcriptase, telomerase, which adds a short, tandemly repeated DNA sequence to chromosome ends for telomere elongation [2,3]. Insects in the order Diptera lack both telomerase and the short terminal repeats found in other organisms. In particular, the telomeres of *Drosophila melanogaster* contain three families of non-long terminal repeat (LTR) retrotransposons, *HeT-A*, *TART*, and *TAHRE* (jointly termed HTT), which transpose specifically to chromosome ends and attach using their 3’ oligo(A) tails [4,5]. Among these three families of elements, *HeT-A* is most abundant, comprising as much as 80–90% of the total number of elements [4,5]. Telomeric chromatin consists of the HTT elements and different proteins that are bound to them [6]. The rate of transposition of these HTT elements may depend on an equilibrium between the level of their expression and the chromatin-bound proteins [5,7,8].

Telomere elongation may also be accomplished by a recombination-based mechanism, including terminal gene conversion using a neighboring telomere as a template [9,10,11]. This mechanism has been also been observed in yeast and in certain human cancers and immortalized mammalian cells, in which overall telomere length increases in the absence of telomerase activity. This recombination-mediated telomere elongation mechanism is called Alternative Lengthening of Telomeres (ALT) [12,13,14] and tends to be most prevalent in tumors of mesenchymal origins [15]. A recent study showed that ALT may also be found in normal mammalian somatic cells [16].

The genes involved in telomere elongation and the mechanisms of elongation are not well studied in *Drosophila*. Two independent studies on *Drosophila* identified the dominant factors *Tel* [17] and *E(tc)* [18], which developed telomeres several-fold longer than controls, to the extent that these differences can be observed microscopically in polytene chromosomes. Mutations in *Su(var)205*, which encodes the HP1a protein, and deficiencies for components of the Ku70-Ku80 complex are also dominant telomere elongation mutations. While *Su(var)205* mutants seem to increase both HTT transposition frequencies and terminal gene conversion, and Ku deficiencies increase gene conversion, the specific mechanisms by which telomere elongation occurs in these mutants are not understood [4,5]. Increased expression of Het-A transcripts and elongated telomeres were also found as a consequence of loss of the *Drosophila* hnRNPA1 homolog, Hrb87F [19], which plays several roles in different processes such as gene expression, organization of the nuclear matrix, and heterochromatin formation. However, whether these effects on telomere elongation are indirect or due to a specific function at chromosome ends is still unclear. Interestingly, the involvement of hnRNPA1 also in telomere regulation of higher eukaryotes [20,21] indicates that it could serve an evolutionarily conserved role at chromosome ends.

Early efforts to map *Tel* [17] allowed meiotic recombination between the *Tel^1^*-bearing chromosome and a multiply marked chromosome; the results located *Tel^1^* at 69 on the genetic map, which translates roughly to 92 on the cytogenetic map. Meiotic mapping of *E(tc)* [18] indicated that this gene is in the same vicinity. In the present study, we took advantage of the observation that there is no meiotic recombination in *Drosophila* males. Thus, site-specific genetic recombination induced by double-strand breaks that result from the excision of DNA transposons can be identified [22]. We used both *P* elements and *Minos* transposons to induce recombination, which allowed the localization of *Tel^1^* to a 15 kb region in the middle of the right arm of chromosome 3 (3R) at 92A. Whole-genome sequencing resulted in the identification of many single-nucleotide polymorphisms (SNPs) and small insertion/deletion polymorphisms (indels) in the *Tel^1^*-bearing genome relative to the reference sequence. Comparison of the *Tel^1^* genomic sequence with a collection of inbred lines of the Drosophila Genetic Reference Panel (DGRP) [23] eliminated all of these SNPs and most of the indels, and mapped *Tel^1^* to a 3 bp deletion (TGT) at 3R:19,366,069-71 in the middle of intron 8 of *Ino80*. Finally, CRISPR/Cas9-induced deletions that removed TGT and surrounding regions exhibited the *Tel^1^* telomere phenotype, confirming that the middle region of *Ino80* intron is indeed required for a proper regulation of telomere length.

## 2. Materials and Methods

### 2.1. Mapping by Site-Specific Recombination

Transposon-induced male recombination was performed as per the mating scheme reported earlier [22,24]. The chromosome carrying the *Tel^1^* mutation was marked with two mutations with eye color phenotypes, *st* and ca. An *st Tel ca* chromosome was made heterozygous in males with a *P* element and the Δ2-3 transposase. Recombinant chromosomes bearing either *st* or *ca* were collected and put into stocks. Stock generation zero occurs at the time the stock was established, two generations after the homozygous recombinants were obtained. These stocks were maintained for a further 12 generations, flies from homozygous recombinants at generation zero, six, nine, and twelve were collected and frozen for DNA isolation in order to assay the *HeT-A* copy number over time.

*Minos* element-induced male recombination mapping is the same as the *P* element-induced male recombination procedure, except that the *P[hsILMiT]* transposase was used in place of Δ2-3. The *P[hsILMiT]2.4* transposase is under the control of a heat shock promoter; therefore, the larvae generated by the cross of heterozygous *st Tel ca*/*Minos* males, were exposed to heat shock at 37 °C in a water bath for 1 hr daily from day two to day six post egg laying [25]. 

### 2.2. Genome Sequencing, Mapping to Reference, and De Novo Assembly

DNA was isolated from approximately 30–40 adult flies by standard procedures of lysis, phenol:chloroform extraction, and ethanol precipitation. The DNA pellet was resuspended in TE buffer (10 mM Tris, 0.1 mM EDTA, pH 7.8). DNA quality and concentration were estimated using a Qubit dsDNA BR Assay Kit and measured by Qubit 2.0 Fluorometer (Life Technologies), as per the manufacturer’s protocol. Five micrograms of DNA were taken for library preparation. 

All the Illumina genome sequence generated for this project can be found in BioProject Accession PRJNA255315 at NCBI. Genome sequencing was done on an Illumina GA IIx sequencer following standard protocols by the NIH Intramural Sequencing Center. For a detailed step-wise protocol for library preparation and genome sequencing, see Appendix A. For *Tel*, 66,654,840 paired reads of 101 bp length were obtained, and for *y w*, 76,757,736 reads were obtained, representing a genome coverage of 48X and 55X, respectively. Genomic reads for each strain were mapped to the *D. melanogaster* reference by two methods. First, all the reads were imported into CLC Genomics Workbench 4.8 and mapped using the parameters Min distance = 150, Max distance = 10,000 to the GenBank-annotated chr3R of dm3 (Release 5 from ftp.ncbi.nih.gov). Second, the same raw reads were mapped to the same reference sequence with bwa 0.6.0 [26] using default settings.

The genomic reads were also assembled de novo by two methods: First, in CLC Genomics Workbench 4.8 using parameters of Min distance = 150, Max distance = 2000, which generated the highest N50 for *Tel* (28.8 kb); Then, in ABySS 1.2.3 [27], a range of kmers from 25–65 was tested using the *Tel* sequence, with a kmer setting of 45 generating the highest N50 (45.3 kb). This setting was subsequently used for both genomic assemblies; all other ABySS settings were default.

### 2.3. Variant Detection 

SNPs and indels (referred to as deletion–insertion polymorphisms, DIPs, by CLC Genomics Workbench) were identified from mapped reads in comparison to the reference genome by two methods: First, using CLC Genomics Workbench 4.8 with the SNP and DIP Detection tools at default settings. Separately, the bwa assemblies were imported into CLC as bam files, and both SNP and indel detection were performed on these assemblies, as above. As this software is not trained for detecting large indels (>5 bp), we scanned a large mapped region of 79 kb (chr3R: 19,325,278–19,404,278) manually and identified additional indels, which had not been detected by the CLC Work bench software (QIAGENE, DK-8000 Aarhus C, Denmark). To complement this SNP/indel analysis by CLC, the same assemblies (CLC and bwa) were also analyzed for SNPs and indels using the pileup program of SAMtools 1.6 [28].

Separately, the contigs spanning the 79 kb region of interest were extracted from each of the de novo assemblies and were aligned to the corresponding reference region with MAFFT 6.849 [29] and manually inspected for indels.

### 2.4. Comparison to DGRP Data 

Files containing the SNPs identified in 162 DGRP [23] lines on chr3R were downloaded (Freeze 1, August 2010 release; http://www.hgsc.bcm.tmc.edu/content/drosophila-genetic-reference-panel (accessed on 3 May 2011)), and the chromosomal coordinates of those 159 strains of normal telomere length were compared with the SNPs identified in the *Tel* genome assembly (SNPs identified by either CLC or pileup). Any SNP identified in the *Tel* genome that was also identified in the SNP collection from DGRP was ruled out as possibly causing the *Tel* phenotype. 

As there were no indel data for DGRP lines in Freeze 1, each indel found in the *Tel* genome assembly, as described above, was compared to the DGRP data. For a subset of eight DGRP lines, the Illumina fastq sequence was downloaded from SRA (SRP000694, Lines 40, 85, 177, 321, 352, 405, 426, 802), imported into CLC Genomics Workbench, and assembled to the chr3R reference, as above, and indel detection was performed. Any indel identified in the *Tel* genome and also found in one or more of these DGRP lines was ruled out as potentially causative. For the indels discovered by manual inspection of both the *Tel* assembly to reference and the *Tel* de novo assemblies, a separate local de novo assembly strategy was used for comparison to a subset of the DGRP population. Using 200–300 bp of reference sequence around a candidate indel as bait, BLAT [30] was used to identify individual reads covering this region from the fasta sequence of a given DGRP line. These reads were then extracted, assembled, and compared to both the *Tel* and reference sequences. Any manually identified indel also found in one or more of these DGRP lines was ruled out as a candidate mutation.

### 2.5. Real-Time PCR

DNA was isolated from 20–30 flies by using DNeasy Blood & Tissue Kit (Qiagen) columns, as per the manufacturer’s protocol. For large numbers of samples, DNA was isolated from 10 flies of each line using Agencourt DNAdvance Genomic DNA Isolation Kit (Beckman Coulter), as per the manufacturer’s protocol. DNA isolation steps were handled by Biomek 4000 Liquid Handling System (Beckman Coulter robotic system). DNA was eluted in 50 ul water. The DNA concentration was estimated by using NanoDrop 2000 (Thermo Fisher Scientific, CA, USA) and diluted to a concentration of 10 ng/µL, using sterile water. 

Primers used for real-time PCR are: 

RpS17-F: 5′AAGCGCATCTGCGAGGAG3′,

RpS17-R: 5′CCTCCTCCTGCAACTTGATG3′, 

HeT-9D4GAG-ORF-F: 5′TTGTCTTCTCCTCCGTCCACC3′,

HeT-9D4GAG-ORF-R: 5′GAGCTGAGATTTTTCTCTATGCTACTG3′.

Predicted sizes of amplicons are 195 bp for RpS17 and 152 bp for the HeT-9 D4 GAG-ORF. GenBank accession number for *HeT-A* element 9D4 is X68130 and for RpS17 is M22142 [31,32]. An aliquot of 20 ng of each DNA sample was taken for quantitative PCR using 50 nM of each primer and 5 µL of 2X Power SYBR green PCR Master Mix (Applied Biosystems) in a 10 µL reaction volume. These samples were amplified under the following conditions: 95 °C for 10 min (polymerase activation), followed by 40 cycles containing denaturation at 95 °C for 15 s, and annealing/extension at 60 °C for 1 min. Real-time PCR was run using ABI Prism 7900 HT Sequence detection system (Applied Biosystems).

Competitive threshold (Ct) values for each sample were collected for *HeT-A* primers (9D4 element GAG ORF) and for control Rps17 (ribosomal protein17) primers. Delta Ct values for each sample were calculated by normalizing *HeT-A* Ct values to control Ct values and graphed using Microsoft Excel. Each DNA sample was run in triplicate to estimate average Ct values.

### 2.6. Generation of Crispr/Cas 9-Induced Deletions

To create CRISPR/Cas 9-induced deletions that include the TGT sequence (3R:19,366,069-71) of the *Ino80* intron-8 region, we introduced two sgRNAs, each containing a protospacer sequence, into the *pCFD4 U6-1 U6-3* tandem gRNAs vector. PAM sequences were selected by using CRISPR Optimal Target Finder online tool (http://targetfinder.flycrispr.neuro.brown.edu/ (accessed on 10 February 2020). Guide sequences were cloned into *pCDF4* using a ligation-independent homology-directed cloning strategy by following CRISPR Fly Design protocol (https://www.crisprflydesign.org/grna-expression-vectors/ (accessed on 14 February 2020)). The following primers were designed:

G1 F:

5’-TATATAGGAAAGATATCCGGGTGAACTTCGGGGAAAGAGGGCAAACGAAG

TTTTAGAGCTAGAAATAGCAAG-3’ containing a 5’ U6-1 promoter complementary region followed by a G-N 19/20 sgRNA sequence (in bold) and a 3’ gRNA core complementary sequence.

G2-R:

5’-ATTTTAACTTGCTATTTCTAGCTCTAAAACTCCCTTTGAGGCCTTTCAACGAC

GTTAAATTGAAAATAGGTC-3’ with a 5’ gRNA core, N19/20 sgRNA sequence (in bold), and 3’ U6-3 promoter reverse and complementary sequences.

Oligos were employed to amplify the *pCDF4* template following the program: initial denaturation at 98 °C for 30 s; 35 cycles of the denaturation step at 98 °C for 10 s, annealing at 66 °C for 15 s, extension at 72 °C for 30 s; final extension at 72 °C for 5 min. After purification with Nucleospin gel and PCR Clean Up Kit (Macherey-Nagel), the PCR product was combined with linear *pCFD4* plasmid, previously digested with BbsI enzyme and gel-purified, in a cloning reaction with Neb Builder HiFi DNA kit. A final *pCDF4* vector containing both guides was sent to BestGene Inc (Chino Hills, CA) and injected into *y v* flies. Transgenic *v^+^* flies bearing the two sgRNAs (*y v*; *gRNAs v+ Ch. II*) were selected and then crossed with *y w*; *nos Cas9/Cyo* flies, which express Cas 9 only in the germline. The resulting *gRNAs/nos Cas9* embryos were injected with the ssDNA template (BiofabResearch, Rome, IT) containing the TGT-specific deletion flanked by homology arms of the corresponding *Ino80* intron-8 region to allow Homologous Recombination-mediated repair. The ssDNA sequence is indicated below:

5′GGCATTTGGTGCTTGGTAGCTTGGTAGAATATTGGGGAAAGAGGGCAAACGAAAGGCCGATTAAATACCAATATGAGTTTTTGGTCAAATTATTCCGGTATGCGCCAAATACATGAACGGCAAATACCTTTGTTCTTGTTGAAAGGCCTCAAAGGGAAGGAGACGAGAATAAGGGGCCACACTCCTTCCAATGGTGTTTGAGAA 3′.

To recover lines bearing mutant third chromosomes, single adults resulting from the ssDNA injection (designed as *A*, *B*, *C*, *D* and *F*) were crossed with *MKRS/TM6B* flies and balanced over *TM6c* by crossing the *TM6B* progeny with the *Ap^xa^/TM6C* strain. For each ssDNA-injected fly, we established at least five independent single-chromatid stocks (indicated as *A1-5*, *B1-5*, *C1-5*, etc…). Total DNA from each homozygous line was extracted, PCR-amplified, and then sequenced.

### 2.7. Chromosome Cytology and Immunostaining

Polytene chromosomes for anti-Hoap immunostaining were prepared as previously described [33] and incubated with rabbit anti-HOAP (1:100). Slides were mounted in Vectashield medium H-1200 with DAPI to stain DNA, and salivary gland preparations were analyzed using a Zeiss Axioplan epifluorescence microscope (CarlZeiss, Oberkochen, Germany), equipped with a cooled CCD (charge-coupled device camera; Photometrics, Woburn, MA, USA). Greyscale digital images were acquired as separate files, which were converted to .psd format, pseudocolored, and merged.

## 3. Results

### 3.1. Transposase-Induced Male Recombination Mapping

As there is no meiotic recombination in *Drosophila* males, it is possible to identify site-specific recombination events generated by transposable elements that transpose by a cut-and-paste mechanism and leave a double-strand DNA break in their wake [22,24]. In general, our procedure was similar to previous work [22,24]; in particular, it involved generating heterozygous *st Tel^1^ ca*/transposon males, inducing transposition with an exogenous transposase, and recovering recombinant *st* or *ca* chromosomes to be tested for the *Tel* phenotype. Initial efforts to map the *Tel^1^* mutation by male recombination were limited by the paucity of useful transposon insertions in the surrounding chromosomal region and continued as new insertion chromosomes became available. The assay used to identify *Tel^1^* on the recombinant chromosomes changed over time. The assay used in early recombination experiments (Table 1, Round 1) used a cytogenetic analysis of heterozygous chromosomes exposed to *Tel^1^* for two years [17]. Later, relative telomere length was estimated by measuring the relative copy number of the open reading frame (ORF) of *HeT-A* at zero, six, nine, and twelve generations after a recombinant stock was established [31,32]. The initial round of mapping, using seven *P* element insertions lying in the 91–93 cytogenetic region (Table 1), showed that *Tel^1^* mapped between the *P[PZ]Dl^05151^* and *P[SUPor-P]CG16718^KG06218^* transposon insertion sites (hereafter, we refer to the transposon insertions simply with their allele designation; full names are listed in Table 1). The physical location is 3R: 19,326,218 to 3R 19,641,774, a region of 316 kb. This region showed a surprising paucity of *P* element insertions.

As new *P* element insertions became available, we used three transposons lying within this 316 kb region: *P[XP]Ino80^d10097^, P[EPgy2]CG31221^EY10678^*, and *P[XP]Dys^d03320^* (Table 1, Round 2; Figure 1A). All three *st*-bearing recombinant chromosomes generated using *P* element insertion *d10097* showed telomere elongation from generation 0 to 12, whereas the three *ca*-bearing recombinant chromosomes from the same *P* element did not show significant telomere elongation (Figure 1B). Thus, *Tel^1^* lies to the left of this *P* element insertion (3R:19,403,413). Similar results were obtained for recombinants from *P* element insertions *EY10678* and *d03320* (see Appendix A), both of which are to the right of *d10097* (Figure 1A). These results mapped *Tel* to 3R: 19,326,218 to 19,403,413, a region of ~77 kb.

*Minos* elements were also used to induce recombination (Table 1), although *Minos* elements had not previously been shown to induce recombination in males. Two *Minos* insertions, *MB02141* and *MB0163*, lying to the right of *d10097* in the 316 kb region showed similar results (see Appendix A), indicating that they are situated to the right of *Tel^1^*, as expected. Two *Minos* insertions in the 77 kb region were selected for further mapping studies (Table 1, Round 3; Figure 1). Even after heat shock, these *Minos* elements generated only a few recombinant males. We obtained only one *st*-bearing recombinant chromosome from each of these *Minos* transposons (Table 1). The *st* recombinants for *MI03112* and *MI02316* showed no evidence of telomere elongation from generation zero through twelve (Figure 1B). This result eliminated the region to the left of *MI02316* as containing *Tel^1^* and mapped *Tel^1^* to a ~15 kb region (3R: 19,388,379 to 3R:19,403,413).

### 3.2. Telomere Length in Transposon Insertion Lines

We measured the relative *HeT-A* copy number in the transposon lines used to induce site-specific recombination. Q-PCR analysis showed that all of the lines, except one bearing *MB09416*, had telomeres comparable in length to the Oregon-R control (Figure 2). The relative *HeT-A* copy number in the *MB09416* insertion stock was highly elevated and similar to that of *Tel^1^*. The *MB09416 Minos* element is inserted at 3R: 19,398,726, which is in intron 8 of *Ino80* and within the 15 kb *Tel^1^* region identified above. UCSC genome browser maps indicate that the insertion site lies in a not well-conserved sequence (Appendix A). As it is likely that the high *HeT-A* copy number in this line might interfere with the ability to observe an increase in telomere length, this transposon was not used in the mapping of *Tel^1^.* It is also possible that the genome of the *MB09416* line carries a genetic factor, either at the insertion site or elsewhere, and that it has a phenotype similar to that of *Tel^1^* and might therefore confound the analysis.

### 3.3. Effect of Tel Copy Number on Telomere Length

Different deficiencies and duplications spanning region 92A3 (Figure 3A), where *Tel* was mapped, were analyzed for an effect on telomere length by measuring the relative *HeT-A* copy number by Q-PCR in the respective stocks. With the exception of *Df(3)BSC475* and *Df(3R)DI-M2,* which induced as yet unclear moderate increase and decrease in *Het-A* genomic copies (*p* < 0.001), respectively, none of the deficiency and duplication stocks showed a substantial accumulation of telomeric HeT-A copy number (Figure 3B). Thus, it appears that neither a 50% increase nor a 50% decrease in the copy number of the region around *Tel* had an effect on telomere length.

### 3.4. SNP and Indel Identification

The genomes of the three strains *Tel^1^, y^1^ w^1^*, and *E(tc)* were sequenced using the Illumina GAIIx platform. As *Tel^1^* appeared in a natural strain (caught in Endine Gaiano, near Bergamo, Italy) that had not been outbred to any laboratory flies [17], there was no useful wild-type control. *E(tc)*, however, appeared in a *y^1^ w^1^* laboratory stock [18]; therefore, we used a *y^1^ w^1^* stock as a wild-type control. Upon sequencing, the *E(tc)* genome appeared to be highly heterozygous. We therefore assumed the stock was contaminated and did not pursue it further. The other two stocks were sequenced to 48X and 55X coverage, respectively. Both genomes were assembled to reference using CLC Genomics Workbench and bwa. In addition, two de novo assemblies, using CLC Genomics Workbench and ABySS, were generated. Concurrently with the *Minos* transposon recombination mapping, a 79 kb genomic region of 3R: 19,325,278 to 19,404,278, roughly the region between inserts *05151* and *d10097*, extended slightly on either end, was analyzed for SNP and indel variations with the above assemblies. Within this region, there were 626 SNPs and 88 indels identified on the *Tel^1^* chromosome compared to the reference using the CLC Genomics SNP and DIP Detection analysis (Table 2). A similar number of variations (586 SNPs and 80 indels) were also found in the *y^1^ w^1^* control strain. After eliminating common variations between *Tel^1^* and *y^1^ w^1^* in this region, we are left with 332 SNPs and 53 indels that appear to be unique to the *Tel^1^* genome in this 79 kb region.

Current variant calling tools are only proficient at defining small indels (1–5 bp) [34,35]. To detect larger indels, we scanned *Tel^1^* genomic assemblies manually and detected 13 polymorphisms of 5 bp or larger (Table 2). These large indels, present in the *Tel^1^* genome but not in *y^1^ w^1^*, were analyzed by PCR with primers flanking these indels and by Sanger sequencing of PCR products (Appendix A). A limitation of the assembly to reference strategy for variant identification is that potential novel insertions that are not present in the reference sequence are not detected by this approach. To search for such variations, we aligned the de novo assemblies of the *Tel^1^* genome to the reference. Manually scanning this alignment identified an additional 14 insertions not found by the above methods (Table 2).

### 3.5. Comparison of Variations to DGRP Data

To differentiate natural polymorphisms among these remaining SNP and indel variations found in the *Tel^1^* genome, we compared them with the genomes available from the DGRP [23], a collection of wild-caught, inbred *Drosophila* strains whose genomes have been sequenced. Our hypothesis is that, if any variant found in the *Tel^1^* genome is also found in a DGRP line with normal-length telomeres, that variant can be ruled out as causing the *Tel* phenotype. As a first step, all DGRP lines were tested for the relative *HeT-A* copy number as a proxy for telomere length. The *HeT-A* copy number data for these strains fit a log-normal distribution, with three outliers that had copy numbers higher than the *Tel^1^* strain (Figure 4). These three lines, RAL-161, -703, and -882, have been excluded from the following discussion and are described elsewhere [32].

As the remaining DGRP lines have what we consider to be normal telomere length, close to that found in the Oregon-R control, the SNPs identified from Freeze1 (August 2010) of the DGRP lines [23] were compared to the SNPs in the *Tel^1^* genome identified by the CLC SNP detection software. All the SNPs found in the 79 kb region around *Tel^1^* were also found in the DGRP lines (Table 2). Thus, all the SNPs found in the *Tel^1^* genome are natural polymorphisms with little expected effect on telomere elongation. No indel data were available for DGRP lines in Freeze1. We therefore identified indels in a selection of eight DGRP lines for the 79 kb region of interest. All except two indels found on the *Tel^1^* chromosome, a deletion of C at 3R:19,352,437, and a deletion of TGT at 3R:19,396,067-69 were also found in one or more lines from the DGRP collection (Table 2). The deletion of C is located in a large intergenic region, 11 kb to the right of *Dl* and more than 16 kb to the left of *CG43203*, while the deletion of TGT is located in intron 8 of *Ino80*. The latter is the only indel-specific to the *Tel^1^* genome in the 15 kb region of interest and therefore was identified as the *Tel^1^* mutation.

### 3.6. Comparison to modENCODE Data

RNA sequence coverage for the 15 kb *Tel* region was analyzed by comparison with the modENCODE database, including stage and tissue-specific transcript expression levels. This analysis shows that the candidate *Tel^1^* mutation was not included in a transcript at any stage in any tissue (Figure 5) and suggests that *Tel^1^* could be acting to alter the expression of other transcripts near or within the *Ino80* locus. The UCSC Genome browser map for this 15 kb region was examined for sequence conservation among *Drosophila* species and other insect species. This analysis shows that, even though the candidate *Tel^1^* mutation is noncoding, it is in a well-conserved region, similar in the level of conservation to neighboring coding regions (see Appendix A).

### 3.7. Transcript Analysis

The transcript levels from the ovaries of *Tel^1^* and Oregon-R were analyzed for nine genes in the vicinity of the *Tel^1^* mutation (TGT deletion), which include *Ino80* and those found within its introns. Quantitative PCR with cDNA from these two lines showed that there is no significant difference (*p* > 0.05) in the expression levels between the two strains for most of these genes (Figure 6). The transcript level of *Ino80* around the exon 8–9 junction that spans the intron 8, where *Tel^1^* mutation is located, also showed an intact transcript with normal expression similar to other parts of the transcript, indicating that the *Tel^1^* mutation does not interfere with local splicing (Figure 6). *CG18493*, however, showed a 15-fold lower expression in *Tel^1^* compared to the control (*p* = 0.0006), and *CG3734* showed a slight reduction in expression in *Tel^1^* ovaries (*p* = 0.0103). Given that a 50% reduction in the *Tel^+^* copy number appears to have no effect on telomere length (Figure 3), that after a Bonferroni correction the small decrease seen for *CG3734* expression (Figure 6) may not be considered significant, and the expression of *CG18493* was not statistically reduced in CRISPR/Cas9-induced *Tel* alleles described below, it seems likely that expression of both *CG18493* and *CG3734* genes is not relevant for the *Tel* phenotype.

### 3.8. Generation of CRISPR/Cas 9-Induced Tel1 Deletion Alleles

To ask whether a deletion of the TGT sequence in the *Ino80* intron 8 yields a *Tel* phenotype, we sought to use CRISPR/Cas9-mediated homology repair to induce the 3 nt deletion, as well as other small deletions encompassing TGT, and check whether these deletions elicited elongated and/or fused telomeres. By using a 205bp ssDNA donor as the template bearing the TGT deletion (see Methods), we obtained 22 potential viable deletions that were established as independent stocks by September 2020. Sequence analysis revealed that these lines identified 10 distinctive small deletions of different sizes ranging from 6 to 222 bps, but none of them identified the 3 nt TGT deletion only (Appendix A). We also noticed that while seven deletions (namely Δ*A3*, Δ*A4*, Δ*B5*, Δ*C11*, Δ*D5*, Δ*F2*, and Δ*F5*) uncovered the TGT sequence, the Δ*C6*, Δ*C10*, and Δ*G3* deletions removed small sequences of 12, 11, and 22 nt, respectively, adjacent to the expected 5′ break site, without including TGT (Appendix A). Moreover, whereas the 3′ junction sequences of all deletions and the 5′ junction sequences of Δ*C4*, Δ*F5*, and Δ*G3* were “clean” breaks, the 5′ breakpoint regions of the remaining deletions contained either putative microhomology sequences (Δ*A4*, Δ*C10*, and Δ*F2*) or stretches of AATATTGG repeats that, in the case of Δ*A3*, Δ*A4*, and Δ*D5*, replaced the TGT-containing region (Appendix A). We then asked whether any of these deficiency-bearing lines exhibited a *Tel* phenotype. Three months after the establishment of stocks bearing the deletions (December 2020), by immunostaining for the telomeric specific marker HOAP [36], we found that polytene chromosomes from heterozygotes for each deletion elicited neither elongated telomeres nor telomere associations, which are normally diagnostic of the *Tel^1^* phenotype [17]. However, when we repeated the cytological characterization after 6 months (March, 2021), we observed that, in the same heterozygotes, telomeres were longer than the wild-type and underwent fusions as expected for *Tel^1^* mutants (Figure 7). Finally, our Q-PCR analysis revealed that these deletion-bearing lines also show a statistically significant increase in the HeT-A copy number after 6 months, thus confirming that these lines represent bona fide new CRISPR/Cas 9-induced *Tel* alleles (Figure 7C). Interestingly, the finding that deletions which do not enclose TGT also display elongated and fused telomeres indicates that a proper *Tel* function requires intact regions within the *Ino80* intron beyond TGT. Thus, although our CRISPR/Cas 9 approach failed to induce a specific 3nt TGT deletion, it allowed the identification of additional sequences in the *Ino80* intron 8 that are required for the regulation of telomere length.

## 4. Discussion

### 4.1. Mechanism of Telomere Elongation

The stability of telomeres in *Drosophila* depends on terminin and non-terminin telomeric proteins [9,37]. The terminin proteins Moi, Ver, HipHop, and HOAP are found only at telomeres, whereas non-terminin proteins HP1, the ATM, and ATR kinases, and the proteins of MRN complex also have biological roles apart from their involvement in telomere maintenance and structure [9,37]. Mutations in any of the genes encoding these proteins cause telomere fusions and abnormal cell divisions. However, only mutations in the HP1-encoding gene *Su(var)205* [38,39], *Hrb87F* [19], and the Ku70/Ku80 complex are associated with telomere elongation. The exact mechanism and the genes involved in telomere length homeostasis in *Drosophila* are largely unknown. There are reports of RNAi control over HeT-A, TART, and TAHRE transcript levels (reviewed in [5]), but the exact mechanism of its involvement in telomere length homeostasis is not well-characterized. Two dominant mutations, *Tel* and *E(tc)*, showed telomere elongation [17,18]. However, whereas the *E(tc)* mutation is associated with elevated rates of gene conversion in telomeric regions, the *Tel* mutation was previously associated with both transposition of the telomeric retrotransposons and gene conversion [40] but not with the transcription of telomeric retrotransposons [41]. The data thus suggest the involvement of *Tel* in a recombination pathway. Interestingly, the original *Tel^1^* mutation and our CRISPR/Cas 9-induced *Tel* deletion alleles have no known phenotype other than telomere elongation and end-to-end attachment in polytene chromosomes [17,42] (this work). Thus, the understanding of the molecular mechanisms that underpin the *Tel* phenotype and the identification of a candidate *Tel* gene will provide us fundamental insights into how *Drosophila* regulate telomere length independently of telomere capping and maintenance. Moreover, the molecular characterization of *Tel* could also contribute to the understanding of recombination-mediated telomere maintenance mechanisms, such as the ALT pathway found in some human cancer cells. Although the ALT pathway is a favorite mechanism of telomere maintenance for some human cancers, the molecular details remain still unknown [12,43].

### 4.2. Mapping Tel Using Transposon-Induced Male Recombination

The *Tel^1^* mutation in *Drosophila* was previously localized by meiotic recombination to 69 on the genetic map [17]. Meiotic recombination in *Drosophila* occurs only in females, but it is possible to induce site-specific recombination in males using transposons and use this for mapping [22,44]. A collection of more than 15,000 publicly available P element insertions means that, in many regions, a resolution of 5–10 kb is possible for P element-induced recombination [45,46].

One major drawback of the P element, however, is its strong bias for insertion into some genes (hot spots) and against insertion into others (cold spots). The region around *Tel* is a cold spot for P element insertions. *Minos*, a member of the *Tc1/Mariner* family of transposable elements, is active in diverse organisms and cultured cells; it produces stable integrants in the germ line of several insect species, in the mouse, and in human cells [25]. To expand the usefulness of transposon mapping in *Drosophila*, collections of other transposable elements with different insertional specificities, such as *Minos* [47,48], have been introduced [46]. *Minos* elements were found to exhibit a generally uniform distribution in the genome [49]. We used available *Minos* elements to refine our mapping of the *Tel^1^* mutation and show for the first time that these transposons can induce recombination events useful for this purpose. This approach localized *Tel^1^* to a region of 15 kb.

### 4.3. Genome Sequencing and DGRP Resources for Tel^1^ Mapping

The molecular lesion associated with *Tel^1^* was identified by deep sequencing of the *Tel^1^* genome and analyzing this sequence for novel SNP and indel variants not found in the DGRP lines [23]. After comparing the variants in the genome bearing *Tel^1^* with DGRP polymorphisms, we ruled out all SNPs and all but one indel in intron 8 of *Ino80* as candidates for *Tel^1^*. Thus, the combination of formal genetics and next-generation sequencing resulted in the identification of the molecular defect in *Tel* as a 3 bp deletion (TGT) at 3R:19,396,067-69. To our knowledge, this is the first study using the DGRP collection to map a Mendelian trait in *D. melanogaster*. The *Tel^1^* mutant used in this study was caught near Endine Gaiano, Bergamo, in northern Italy, likely prior to 1946 [17]. It is of interest that the natural genetic diversity captured by DGRP in a Raleigh, North Carolina, population was of sufficient diversity to identify all of the SNPs and all but two indels within our 79 kb region defined by the transposon mapping. This suggests that the DGRP is an important general resource for genetic mapping of genes in *Drosophila melanogaster*, even from strains not closely related to the standard reference isolates. CRISPR/Cas9-induced deletions in the intronic region encompassing TGT also resulted in a *Tel*-specific phenotype, confirming that intron 8 of *Ino80* contains the *Tel* locus. Telomere elongation was evident 6 months, but not 3 months, after the induction of deletions. Consistent with our previous characterization [17], we speculate that this difference in telomere length depends on the progressive accumulation of HTT genomic copies through recombination and/or gene conversion events over time. The *Drosophila* Ino80 is the ortholog of the human INO80 ATPase, a member of the SNF2 family of ATPases that functions as an integral component of a multisubunit ATP-dependent chromatin remodeling complex [50]. In flies, Ino80 is involved in the regulation of homeotic gene expression and regression of ecdysone-dependent transcription [51,52]. However, our genetic and transcript analyses exclude the possibility that *Tel^1^* is an *Ino80* mutant allele that specifically affects telomere length maintenance. The reason why the TGT deletion and all additional deletions surrounding TGT recovered by CRISPR/Cas9 yielded elongated telomeres still remains unclear. We can speculate that *Tel* specifies a regulatory element in the intron 8 of *Ino80* required for telomere homeostasis. One possibility is that it may encode a still unannotated ncRNA (i.e., microRNA), which could influence the expression of genes required for the maintenance of chromosome end length. Moreover, this sequence may act as a binding platform for trans-acting factors, which are required for the activity of genes that prevent recombination events at telomeres. Although these genes are not known, we can rule out that the genes located in the intron 8 are potential candidates, as their expression is not affected by *Tel^1^* mutations. Alternatively, we can envisage that the TGT-containing sequence could regulate appropriate transcript maturation (i.e., working as a splicing enhancer) and that its loss could affect normal splicing of adjacent exons and generate an aberrant gain of function Ino80 isoform with a dominant effect. Whatever the hypotheses, further genetic and transcriptomic studies are necessary to reveal the entire sequence that identifies *Tel* and the genes regulated by *Tel*.

## Figures and Tables

**Figure 1 cells-11-03484-f001:**
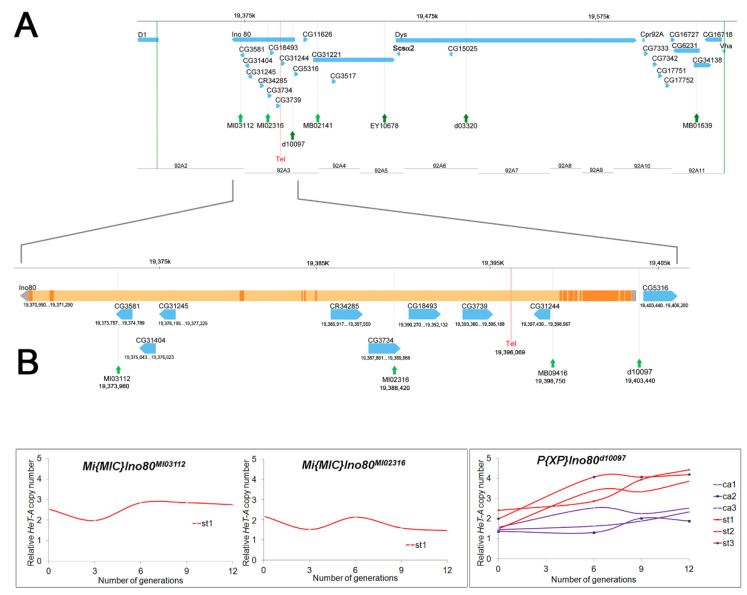
**Localization of *Tel^1^* by site-specific recombination.** (**A**) The upper chromosome map shows the candidate genes between two P element insertion sites, *05151* and *05113* (vertical green lines). This region was identified as containing *Tel^1^* based on Table 1, round 1. Positions of transposons used for further mapping are indicated by green arrows. The *Tel^1^* mutation is boxed in red. The lower chromosome map shows expansion of the 92A3 region. (**B**) Graphs showing the change in relative *HeT-A* copy number (telomere length) in recombinants of *Tel^1^/MI02316*, *Tel^1^/MI03112*, and *Tel^1^/d10097* over 12 generations. The *st* recombinants are shown as red lines; *ca* recombinants as purple lines. These data delimit *Tel^1^* to a 15 kb between inserts *MI02316* and *d10097* (shown as red rectangles in (**A**)).

**Figure 2 cells-11-03484-f002:**
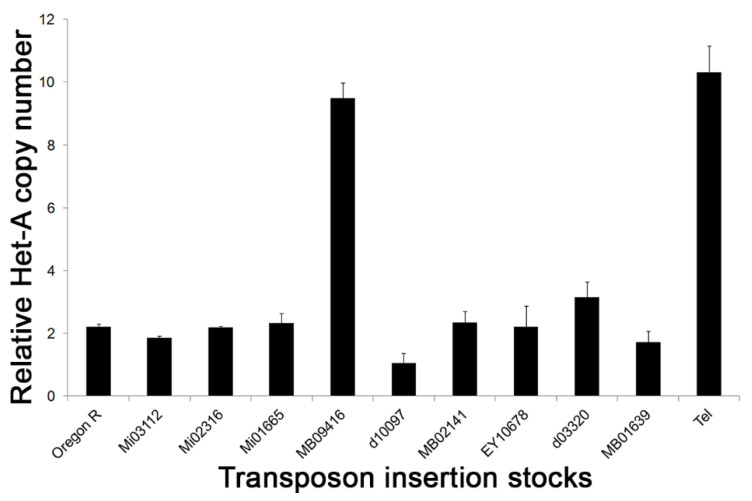
**Telomere length in transposon insertion stocks.** Q-PCR analysis of *HeT-A* copy number in different transposon insertion stocks used for mapping *Tel^1^* mutation. Error bars represent standard deviation measured from the triplicate Q-PCR results. *MB09416* was not used for subsequent site-specific recombination mapping.

**Figure 3 cells-11-03484-f003:**
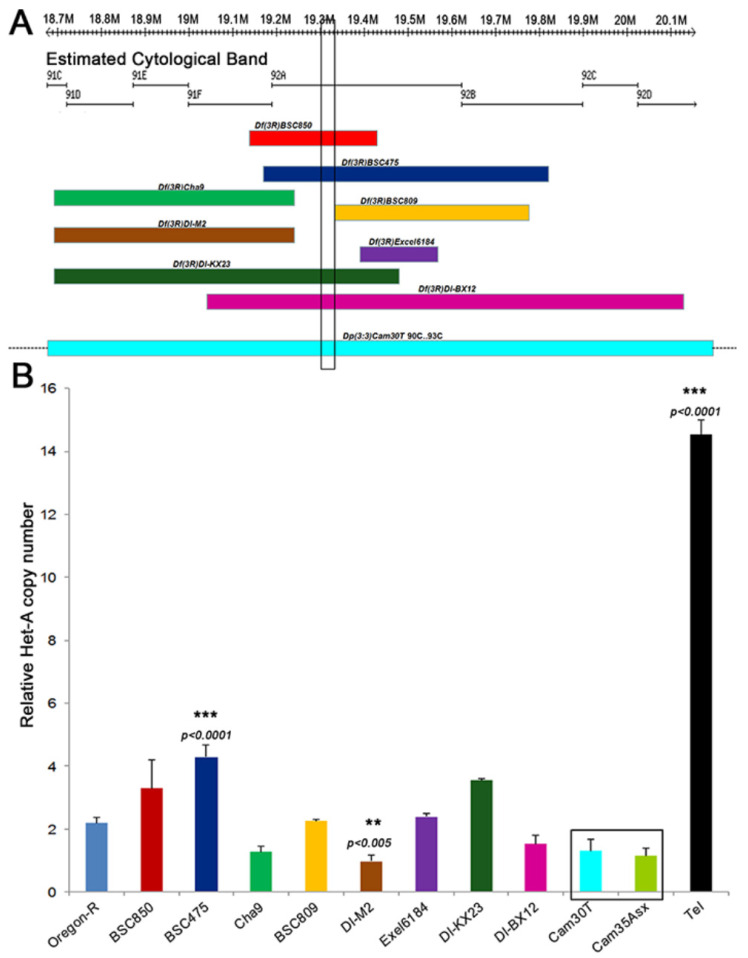
**Dosage effects of Tel on *HeT-A* copy number.** (**A**) The cytogenetic and physical map of genomic region 91C-91D is displayed, highlighting the 15 kb *Tel^1^* region as a vertical box. The lower part of the panel shows the extent of chromosome deficiencies. The bottom rectangle, *Dp(3:3)cam30T*, is a duplication for this region. Dotted lines beyond this rectangle show that the duplication extends beyond the represented region. (**B**) Shows the relative *HeT-A* copy number in stocks of the aberrations shown in A. The highlighted box represents a duplication that includes mapped region (*Dp(3;3)cam30T* covers 90C-93C) and another duplication of a neighboring region of the genome *(Dp(3;3)cam35* covers 67C5-69A5) as a control. The mean from three replicates is represented here and error bars represent standard deviation. ** *p* < 0.005; *** *p* < 0.0001, one-way ANOVA with Sidak correction.

**Figure 4 cells-11-03484-f004:**
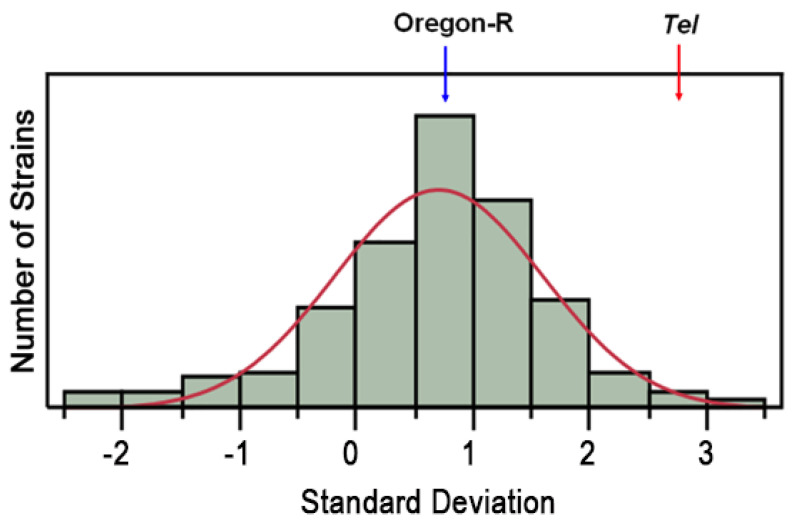
**Telomere lengths in DGRP lines.** A bar graph shows the log-normal distribution of telomere lengths among the 162 DGRP lines measured. The blue arrow shows the position of the Oregon-R control, and the red arrow shows the position of *Tel^1^*. Three lines have *Het-A* copy numbers that exceed three standard deviations from the mean. These are RAL-161, -703, and -882. The red curve indicates the expected distribution.

**Figure 5 cells-11-03484-f005:**
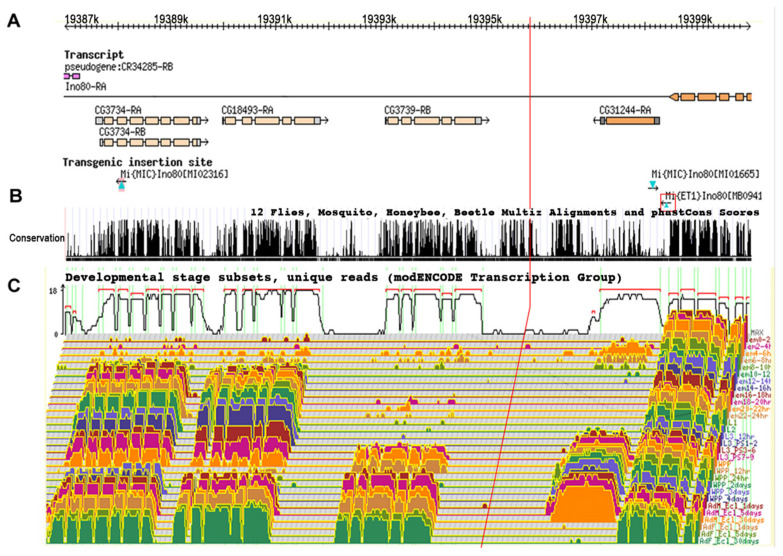
**Genetic activity and conservation of the 15 kb Tel1 region.** (**A**) The genes found in this region are aligned with molecular coordinates. *Minos* insertions used for mapping are shown in cyan triangles. *Minos* insertion M*B09416*, which showed telomere elongation, is highlighted in the red square. (**B**) The University of California Santa Cruz genome browser map highlights sequence conservation in this region among different insect species. (**C**) A developmental transcriptome analysis for the same region as determined by the modENCODE project is also shown. The red vertical line spanning all three panels indicates the position of the TGT deletion.

**Figure 6 cells-11-03484-f006:**
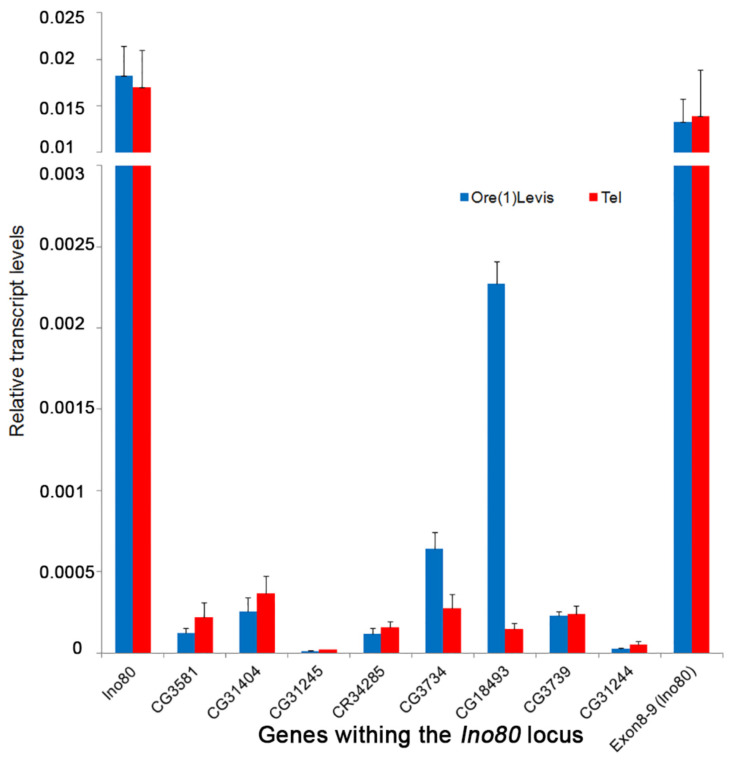
Transcript analysis in the genes near *Tel^1^*. The histogram represents the relative transcript levels from *Tel^1^* mutant (red) and Oregon-R (blue) adult ovaries. The genes analyzed are found near Tel1 (TGT mutation), which include Ino80 and the genes within its introns. The last bar represents the expression of Ino80 gene spanning exons 8–9, around the position of the TGT deletion in intron 8. The expression levels of *CG31244* and *CG31245* are very low, similar to the modENCODE data.

**Figure 7 cells-11-03484-f007:**
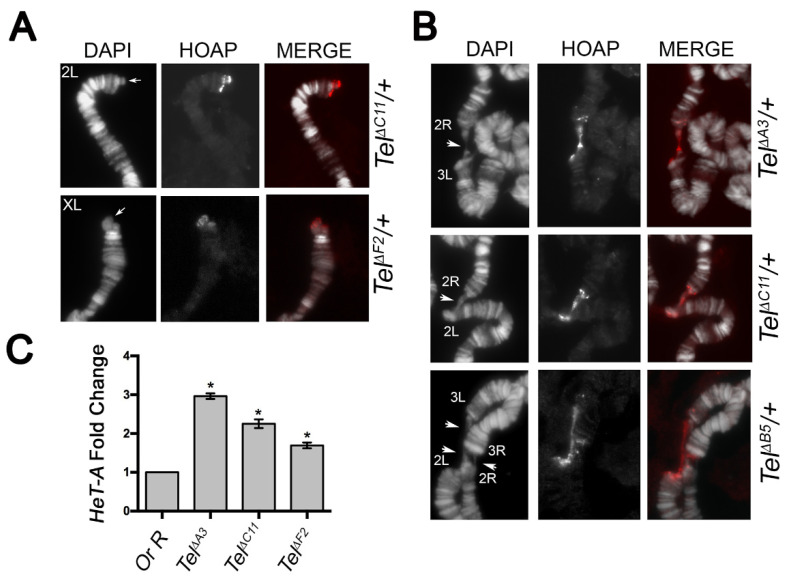
**Telomere elongation in the CRISPR/Cas 9-induced Tel deletion alleles.** (**A**) anti-HOAP immunostaining on 2L and XL chromosome tips from *Tel*^∆*F2*/+^ and *Tel*^∆*C11*/+^ hemyzygotes, respectively, showing protruding telomeric DNA only on the *Tel* mutant chromosomes (arrows). Note that, as expected, *Tel* mutant elongated telomeres do not impair HOAP localization. (**B**) Examples of telomere fusions involving either two or four chromosome tips (arrowheads) from different *Tel*∆ hemizygotes. (**C**) qPCR analysis on third instar larval DNA from two representative *Tel*∆ deletion alleles showing a robust increase in Het-A copy number compared to control (OR-R). Note that *Tel*^∆*B5*^ bears a deletion that uncovers TGT, while *Tel*^∆*C10*^ does not include TGT. * (*p* < 0.05).

**Table 1 cells-11-03484-t001:** Recombinants obtained from each transposon used for site-specific recombination.

Round ^a^	Transposon Insertion	Cytology ^b^	Coordinate ^c^	No. Recombinants	*Tel^1^* Position ^d^
1	*P{PZ}sqz^02102^*	91F4	19,165,876	3	Right
1	*P{lacW}vib^j5A6^*	91F12	19,226,365	6	Right
1	*P{PZ}l(3)05820^05820^*	91F12	19,229,496	3	Right
1	*P{PZ}Dl^05151^*	92A2	19,326,218	2	Right
3	*Mi{MIC}Ino80^MI03112^*	92A3	19,373,898	1	Right
3	*Mi{MIC}Ino80^MI02316^*	92A3	19,388,379	1	Right
2	*P{XP}Ino80^d10097^*	92A3	19,403,413	6	Left
2	*Mi{ET1}CG31221^MB02141^*	92A3	19,414,147	5	Left
2	*P{EPgy2}CG31221^EY10678^*	92A5	19,453,650	4	Left
2	*P{XP}Dys^d03320^*	92A6	19,498,929	3	Left
2	*MI{ET1}CG6231^MB01639^*	92A11	19,629,652	6	Left
1	*P{SUPor-P}CG16718^KG06218^*	92A11	19,641,774	3	Left
1	*P{PZ}Vha13^05113^*	92A11	19,644,018–196,44,026	3	Left
1	*P{hsneo}l(3)neo50^1^*	92B3	19,836,871	3	Left

^a^ Transposons were used to induce recombination as they became available. Succeeding rounds used slightly different procedures, as described in the text. ^b^ Estimated cytological band as reported in FlyBase. ^c^ Nucleotide position in the genomic sequence of chromosome arm *3*R. ^d^ The position of *Tel^1^* relative to the insertion site.

**Table 2 cells-11-03484-t002:** SNPs and indels found in the *Tel1* genome relative to the standard reference sequence of chromosome arm 3R between coordinates 19,325,278 and 19,404,278.

Polymorphisms	SNPs	Indels
Identified by CLC-Genomics	626	88
Identified by manual comparison	-	13
Identified by *de novo* assembly	-	14
**Total**	**626**	**115**
Not in DGRP ^a^	0	2

^a^ There were 159 DGRP lines used in the SNP comparison (Freeze1 data) and eight used in the indel comparison.

## Data Availability

All Illumina genome sequence generated for this project can be found in BioProject Accession PRJNA255315 at NCBI. All Appendix A are available in the Supplementary Information. The data that support the findings of this study are available from the corresponding author upon reasonable request.

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
