# Peer review of "Identification of the Telomere elongation Mutation in Drosophila"

_cells, 2022, doi:10.3390/cells11213484_

Round 1

Reviewer 1 Report

In this manuscript the authors described the identification of a new gene involved in the regulation of telomere elongation in drosophila. This work is the follow up of their previous study describing the isolation of a dominant mutation Tel1 that caused several fold elongation of telomeres. Through the use of P elements and Minos transposons in male drosophila (that do not undergo meiotic recombination), they located Tel1 in a 15kb region (3R: 19,388,379 to 3R:19,403,413). They then sequenced a region including the region of interest and identified a number of SNPs and Indels which after comparison with 159 DGRP lines where reduced to two indels and then to only one (TGT at 3R:19,396,067-69) located in the intron 8 of ino80 within the 15kb region identified using the transposons. They then consider this mutation as the candidate Tel1 mutation. Through comparison with the modENCODE database, they found that the candidate Tel1 mutation is not found in any transcript at any stage in any tissue suggesting that the elongation phenotype is not a direct consequence of gene expression/splicing. They further show that the candidate Tel1 mutation is as conserved as neighboring coding sequence in different drosophila species and other insects. They went on in the mutation of this candidate Tel1 mutation using the CRISPR/cas9 system. Although they failed to obtain the exact deletion of the 3 nucleotides identified in ino80 intron 8, they obtain several mutants displaying small deletions uncovering the TGT sequence. Analysis of these mutants revealed that these mutations lead to the increase of the telomere length after 6 months (but not 3 months).

This is an interestingly and well conducted study that allows the identification of a mutation involved in the regulation of the telomere length in drosophila. Although, it is not clear how this mutation leads to the observed phenotype, this study represents a step forward in the field. I would suggest few modifications of this manuscript to make the manuscript clearer:

-        Point 1: on figure 3, would it be possible to add statistical analysis to determine whether the relative Het-A copy number is significantly different between the different lines in particular as compared to the control Oregon-R. Indeed Cha9 and DI-M2 seem to express less Het-A and could be significant.

-        Point 2: it would be interesting to have the Het-A Q-PCR data of the CRISPR/cas9 clones at 3 months, to see whether there is an increase in telomere size at this scale of analysis.

-        Point 3: it would be interesting to discuss the difference of phenotypes of the CRISPR/cas9 at 3 and 6 months.

-        Point4: In the discussion part, the authors suggest “one possibility…chromosome end length”. Is it compatible with the fact they do not detect this mutation in any of the transcriptomic data they have analyzed? Can the authors envisage RT-PCR and sequencing of this region to validate their hypothesis?

Author Response

See the attached "Response to Reviewer 1" pdf file

Reviewer 2 Report

The length of the telomeres in Drosophila melanogaster is maintained by specialised retrotransposons HeT-A, TART, and TAHRE. The length of the telomeres relies on the number of copies of these transposable elements and several genes, which function is still poorly understood. The authors previously reported a dominant mutation, Tel1, causing elongation of telomeres. In this study, they identified a small deletion within an intron in the Ino80 as the possible molecular basis of the Tel1 mutation. CRISPR/Cas9-induced deletions ratified the requirement of this intron sequence for appropriate regulation of the D. melanogaster telomere length.

Comments and criticisms:

·       In line 324, the MB09416 Minos element at 3R: 19,398,726 shows a highly elevated HeT-A copy number like Tel1. This transposon is inserted upstream of CG31244. Why haven't the authors molecularly characterised this allele or tested genetic interactions with Tel1 or transcriptomic changes? 

·       The effects of deficiencies and the Tel1 and CRISPR/Cas9-induced deletions are seemingly contradictory. The authors show that Tel1 is associated with a small deletion of 3 nucleotides. The CRISPR/Cas9-induced deletions are also dominant mutations causing show telomere elongation defects. However, none of the deficiency stocks uncovering intron 8 of Ino80 showed longer telomeres in Figure 3B. Is that correct? Please provide further arguments on how this can be explained. A non-coding RNA gene seems unlikely if the deficiencies produce no effect. Such RNA should be detected in their RNAseq data.

·       The authors should present more data exploring the potential mechanism. For example, they could have generated transgenic flies carrying the regions with the deletion to test whether they can dominantly cause telomere elongation when the region is in trans. 

·       It would also be interesting to test the influence on the telomere length of trans-heterozygous Tel1, the de novo generated CRISPR/Cas9 alleles, and the appropriate deficiencies. Does Tel1 interact with the MB09416 Mino's mutation? What is the consequence of trans-heterozygous carrying the CRISPR deletions over a deficiency?  

·       Like the MINOS mutation, I found that the lines from the DGRP, RAL-161, -703, and -882, with high HeT-A copy numbers, are of interest to complement the search for possible molecular candidates regulated by Tel1. Do the authors know whether these lines carry SNPs near the Tel1 region? 

·       I have questions about the experiment of transcript levels comparison from Tel1 and Oregon R ovaries. Is Oregon R the right control? Is the ovary the appropriate tissue (gender) to perform the experiments? Why?

·       The Tel1 strain has an uncertain genetic background, so it is not as useful as the newly generated CRISPR strains. Authors should present data and arguments using CRISPR alleles with more controlled genetic backgrounds, preferably males and females, not just ovaries, and perhaps perform a global search rather than restrict the search to changes in the vicinity of the Tel1 deletion.

·       Before ruling out candidates CG18493 and CG3734, shouldn’t the authors perform functional tests with alleles or RNAi? Several reasons may explain that deficiencies do not affect telomere length. One is that the deficiency may not result in a 50% reduction in transcription if another paralog compensates for gene expression. This is not uncommon to happen. In contrast, the Tel1 deletion may structurally affect a regulatory sequence of an RNA with a dominant, neomorphic effect that cannot be mimicked with a large deficiency.

·       In their RNA-seq data, have the authors observed changes in Su(var)205, Hrb87F or the Ku70/ku80 complex? Have they done epistatic studies with these other mutants and their CRISPR/Cas9-induced alleles?

·       The Tel1 strain has an uncertain genetic background, so it is not as useful as the newly generated CRISPR strains. Authors should present data and arguments using CRISPR alleles with more controlled genetic backgrounds, preferably males and females, not just ovaries.

·       Have they sequenced intron 8 of Ino80 from the Minos element strain with elongated telomeres? This mutant appears promising for advancing the molecular basis of the Tel1/intron sequence.

·       A key experiment is to clone that region or all of intron 8 with the deletion of Tel1 or one of its CRISPR alleles and generate transgenes to test the effect of the deletion on trans. This experiment would provide much information and clarify one of the suggestions that the deletion affects an RNA not yet described.

In summary, the studies carried out are well-designed and executed and are long and complex experiments; the result they yield is, unfortunately, a very small advance, which sheds little insight on the nature of the mutation (regulatory sequences, structural variant, a silencer region, RNA, ...) or the mechanism. 

Author Response

See the attached "Response to Reviewer 2" pdf file

Round 2

Reviewer 2 Report

The authors have satisfactorily and convincingly responded to all my previous questions and comments, so there are no issues left. Understandably, some of the experiments may take too long even though they are interesting for future studies.